

# Beryllium-9 in cluster effective field theory

Elena Filandri[1,2]*, Paolo Andreatta[1], Carlo A. Manzata[1],
Chen Ji[3], W. Leidemann[1,2] and G. Orlandini[1,2]

**1** Università di Trento, 38123 Trento, Italy
**2** INFN-TIFPA Trento Institute of Fundamental Physics and Applications,
Via Sommarive, 14, 38123 Povo-Trento, Italy
**3** Key Laboratory of Quark and Lepton Physics (MOE) and Institute of Particle Physics,
Central China Normal University, Wuhan 430079, China

* elena.filandri@unitn.it

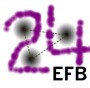

## Abstract

**We study the $^9$Be ground-state energy with non-local $\alpha n$ and $\alpha\alpha$ potentials derived from Cluster Effective Field Theory. The short-distance dependence of the interaction is regulated with a momentum cutoff. The potential parameters are fitted to reproduce the scattering length and effective range. We implement such potential models in a Non-Symmetrized Hyperspherical Harmonics (NSHH) code in momentum space. In addition we calculate ground-state energies of various alpha nuclei. Work is in progress on a calculation of the photodisintegration of $^9$Be with the Lorentz Integral Transform (LIT) method.**

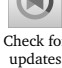
## 1 Introduction

The idea of alpha clustering has a long history, that goes back to the 1930s [1]. By observing alpha decay from nuclei, physicists speculated that they are made up of alpha particles. Nowadays there is much experimental evidence for alpha clustering in nuclei. We mention for example, the $^8$Be decay in two alpha particles, the observation of $^{12}$C Hoyle state as well as the observation of other systems [1] predicted by the Ikeda diagram. Furthermore some of the recent experimental studies strongly support the alpha cluster structure in $^{56}$Ni [2] and in the ground state of $^{40}$Ca [3].

In this context our purpose is to describe these cluster nuclei and some reactions of astrophysical interest, specializing in low energies, where clusters of nucleons behave coherently. In this work we focus on the Borromean system provided by the nucleus of $^9$Be which shows a separation of scales at low energy. For $E < 20$ MeV the dynamics describing the cluster

configuration is insensitive to the internal dynamics of the $\alpha$ particles. Therefore, in order to describe this system, we can use a three-body approach with interactions among nucleon and alpha particles. The cluster approach is not new for the study of $^9$Be. The same technique of clustering was employed by Efros *et al.* in [4], where this nucleus is described as an $\alpha\alpha n$ system and a calculation of the ground state has been made using phenomenological local potentials. Within the same three-body approach, another calculation by Casal *et al.* [5] was performed, where also a phenomenological three-body force was introduced.

In this work instead non-local potentials derived from Cluster effective field theory [6–9], with a more solid theoretical background, are used. The final goal of this project will be to employ these potentials in the calculation of the photodisintegration of $^9$Be by the LIT method [10]. This reaction is particularly interesting since the inverse reaction represents an alternative to the triple alpha process in the formation of $^{12}$C in supernovae events. This paper will be organized as follows. In Section 2 we will introduce the Cluster effective field theory and the potentials used, in Sec. 3 we will present our results and in Sec. 4 the conclusions.

## 2 Cluster Effective Field Theory

In nuclear physics in general nucleons are used as effective degree of freedom, however this is not the only possible choice. In particular, many nuclei present the peculiar property for which the probability distribution of the valence neutrons extends well beyond that of the core and they are called halo nuclei. Others have some parts of the system which can be seen as separated subsystems. In this work we study the Borromean system provided by the nucleus of $^9$Be. The energy needed in order to separate the system into the three effective degrees of freedom is $\sim 1.572$ MeV, while the proton separation energy of $^4$He is $S_p(^4\text{He}) \sim 19.813$ MeV. Comparing these two energies values, one can already see a separation of scales, needed for an Effective Field Theory approach [6–9]. The two types of subsystems of $^9$Be are the $\alpha\alpha$ pair and the $\alpha n$ one. The $\alpha\alpha$ interaction is dominated by the $^1S_0$ resonant state, while the $\alpha n$ system has a resonance in the $^2P_{\frac{3}{2}}$ partial wave at low energies.

Another feature required for an effective theory approach is the power counting. For the $n\alpha$ case, from a physical interpretation one would expect that the two scales are given by $M_{lo} = \sqrt{2\mu_{\alpha n}Q_{\alpha decay}(^5\text{He})} \approx 30$ MeV, $M_{hi} = \sqrt{2\mu_{\alpha n}S_p(^4\text{He})} \approx 140$ MeV. The chosen power counting should also reproduce the known resonance of the system at low energy $\sim M_{lo}$, therefore we need to keep the scattering length term and the effective range one to be of the same order at $M_{lo}$ to guarantee a resonance pole in the T-matrix. We adopt the following power counting [9]:

$$\frac{1}{a_1} \sim M_{lo}^2 M_{hi} , \quad r_1 \sim M_{hi} , \tag{1}$$

$a_1$ being the scattering length and $r_1$ the effective range. Hence, using experimental values for $a_1$ and $r_1$, we get $M_{lo} \approx 50$ MeV and $M_{hi} \approx 170$ MeV.

In the $\alpha\alpha$ case we have three different scales of interest $M_{lo} = \sqrt{2\mu_{\alpha\alpha}Q_{\alpha decay}(^8\text{Be})} \approx 20$ MeV, $M_{hi} = \sqrt{2\mu_{\alpha\alpha}S_p(^4\text{He})} \approx 260$ MeV and the Coulomb one $k_C = 4\alpha\mu_{\alpha\alpha}$. In a similar way to the previous case, but with the following power counting [7]

$$a_0 \sim \frac{M_{hi}^2}{M_{lo}^3} , \quad r_0 \sim \frac{1}{3k_C} \sim \frac{1}{M_{hi}} , \tag{2}$$

and using again the experimental values, we obtain $M_{lo} \approx 20$ MeV and $M_{hi} \approx 170$ MeV. Therefore, we perform an EFT expansion up to the effective range order with a precision given in the $\alpha n$ case by $O\left(\frac{M_{lo,\alpha n}}{M_{hi,\alpha n}}\right) \sim 0.3$, while in the $\alpha\alpha$ one has $O\left(\frac{M_{lo,\alpha\alpha}}{M_{hi,\alpha\alpha}}\right) \sim 0.1$. Moreover in order

to evaluate the range of validity of our EFT, we should also consider the breakdown scale of $\alpha\alpha n$ system. Since we consider a three-body problem we have to take the strictest constraint $M_{hi} = min\{M_{hi,\alpha n}, M_{hi,\alpha\alpha}\}$. With the adopted power counting, in the $\alpha n$ interaction case, the scattering length $a_1$ and the effective range $r_1$ contribute to the leading order (LO), there are no contributions at the next-to-leading order (NLO) and the shape parameter $\mathcal{P}_1$ is next-to-next-to leading order (N2LO). In the case of $\alpha\alpha$ interaction $a_0$ and $r_0$ give contributions to the LO, there are no contributions at the NLO and the shape parameter $\mathcal{P}_0$ is of a higher order.

## 2.1 The Potential

At low energies, one can describe the short-range interaction between two particles with a potential in momentum space of the form,

$$V(\boldsymbol{p},\boldsymbol{p}') = \sum_{i,j=0}^{1} p^{2i}\lambda_{ij}p'^{2j}, \tag{3}$$

where $p$ and $p'$ are the two-body relative momenta and we have introduced the matrix

$$\lambda = \begin{pmatrix} \lambda_0 & \lambda_1 \\ \lambda_1 & 0 \end{pmatrix}. \tag{4}$$

One can, in general, expand a potential in partial-wave components by defining

$$V_l(p,p') = \frac{1}{2}\int_{-1}^{1}\langle\boldsymbol{p}|V|\boldsymbol{p}'\rangle P_l(\hat{\boldsymbol{p}}\cdot\hat{\boldsymbol{p}}')d(\hat{\boldsymbol{p}}\cdot\hat{\boldsymbol{p}}'), \tag{5}$$

$$V(\boldsymbol{p},\boldsymbol{p}') = \langle\boldsymbol{p}|V|\boldsymbol{p}'\rangle = \sum_{l=0}^{\infty}(2l+1)V_l(p,p')P_l(\hat{\boldsymbol{p}}\cdot\hat{\boldsymbol{p}}'), \tag{6}$$

where $P_l$ is the $l$-th Legendre polynomial.
In particular, for a potential dominated by a specific partial wave $l$ one has

$$V(\boldsymbol{p},\boldsymbol{p}') = p^l p'^l g(p)g(p')\sum_{i,j=0}^{1} p^{2i}\lambda_{ij}p'^{2j}(2l+1)P_l(\hat{\boldsymbol{p}}\cdot\hat{\boldsymbol{p}}'), \tag{7}$$

where the $\lambda$ matrix is defined as in (4).
The potential $V(\boldsymbol{p},\boldsymbol{p}')$ is modified by introducing the function $g(p)$, which regulates the short-distance dependence of the interaction, such that $g(p=0)=1$ and $g(p\to\infty)=0$. The two indices $i$ and $j$, in principle, could be larger than 1, but we are limiting them in order to get a phase shift expansion up to the effective range order. This leads for the on-shell T-matrix to the following relation

$$\frac{k^{2l+1}}{T_l^{on}(E)} = -\frac{\mu}{2\pi}\left(\frac{1}{\alpha_l} + \frac{1}{2}r_{e,l}k^2 - ik^{2l+1}\right) + \frac{\mu}{\pi}k_c H(\eta) + O(k^3), \tag{8}$$

with the scattering length $\alpha_l$ and the effective range $r_{e,l}$. Above $H(\eta)$ is a function which takes into account the Coulomb effect present in the $\alpha\alpha$ interaction, for real values of $\eta = \frac{k_C}{k}$ it can be expressed as

$$H(\eta) = Re[\Psi(1+\eta)] - \ln\eta + \frac{i}{2\eta}\left(\frac{2\pi\eta}{e^{2\pi\eta}-1}\right) \tag{9}$$

in terms of the digamma function $\Psi(z) = (d/dz)\ln\Gamma(z)$. The partial wave expansion shown here is important in the study of $^9$Be and $^{12}$C nuclei since, as already mentioned, the two interactions have a dominant wave according to the used power counting. Thus one needs to

find in both cases explicit expressions for the coefficients $\lambda_0$ and $\lambda_1$ in terms of the scattering length and effective range, with a dependence on the cutoff $\Lambda$ necessary to take care of the ultraviolet divergences. The coefficients for the potential were found by expanding the Lippmann-Schwinger equation in partial waves in a similar manner to (6):

$$T(\boldsymbol{p},\boldsymbol{p}') = \sum_{l=0}^{\infty}(2l+1)T_l(p,p')P_l(\hat{\boldsymbol{p}}\cdot\hat{\boldsymbol{p}}'), \tag{10}$$

where

$$T_l(p,p') = p^l p'^l g(p)g(p') \sum_{i,j=0}^{1} p^{2i}\tau_{ij}(E)p'^{2j}. \tag{11}$$

What differs between our two cases is how the Lippmann-Schwinger equation is generated. In the $\alpha n$ case the Lippmann-Schwinger equation takes the form

$$T(\boldsymbol{p},\boldsymbol{p}') = V(\boldsymbol{p},\boldsymbol{p}') + \int \frac{d\boldsymbol{q}}{(2\pi)^3}V(\boldsymbol{p},\boldsymbol{q})\frac{1}{E-\frac{q^2}{2\mu_{\alpha n}}+i\epsilon}T(\boldsymbol{q},\boldsymbol{p}'), \tag{12}$$

where $E = k^2/(2\mu_{\alpha n})$. In the $\alpha\alpha$ case, instead, one has to consider also the presence of the long range Coulomb interaction. Then the T-matrix can be separated as follows

$$T(\boldsymbol{p},\boldsymbol{p}') = T_C(\boldsymbol{p},\boldsymbol{p}') + T_{SC}(\boldsymbol{p},\boldsymbol{p}'), \tag{13}$$

where the $T_C(\boldsymbol{p},\boldsymbol{p}')$ is the pure Coulomb one. The latter satisfies the following equation

$$T_{SC}(\boldsymbol{p}',\boldsymbol{p}) = \langle\psi_{\boldsymbol{p}'}^{(-)}|V_S|\psi_{\boldsymbol{p}}^{(+)}\rangle - 2\mu_{\alpha\alpha}\int\frac{d\boldsymbol{p}''}{(2\pi)^3}\langle\psi_{\boldsymbol{p}'}^{(-)}|V_S|\psi_{\boldsymbol{p}''}^{(-)}\rangle\frac{T_{SC}(\boldsymbol{p}'',\boldsymbol{p})}{p^2-k^2+i\epsilon}, \tag{14}$$

where $|\psi_{\boldsymbol{p}}^{(\pm)}\rangle = \left[1+G_C^{(\pm)}V_C\right]|\boldsymbol{p}\rangle$ with $G_C^{(\pm)}$ the Coulomb Green's function.

In a next step the partial wave decomposition as in (7) has been used in order to solve the Lippmann-Schwinger equation. The resulting expressions are expanded in $k^2/\Lambda^2$ and evaluated for the relevant partial waves. After the addition of the necessary cutoff, $g(k) = e^{-\left(\frac{k}{\Lambda}\right)^{2m}}$, $m \in \mathcal{Z}$, one finds in both cases quadratic equations, leading to two sets of solutions for $\lambda_0$ and $\lambda_1$, one with a positive $\lambda_0$ and negative $\lambda_1$ and one with a negative $\lambda_0$ and positive $\lambda_1$. Later we will call the first solution $\lambda_0$ repulsive and the second one $\lambda_0$ attractive, it is worth pointing out that both sets of solutions generate an attractive potential between the particles. In the $\alpha n$ case we choose the set of more natural size. In the $\alpha\alpha$ case, instead, both sets of parameters are of a rather natural size and therefore we study them both.

In Fig. 1 we show the cutoff dependence for the $\alpha n$ phase shift. For cutoffs between 200 and 300 MeV one finds a good agreement with experimental data. In the inset of the figure one sees that the total cross section correctly reproduces the $^2P_{3/2}$ resonance at $E_R = Q_{\alpha\text{decay}}(^5\text{He}) = 0.798$ MeV with a width of 0.648 MeV [12].

In Figure 2 we show our results for the $\alpha\alpha$ phase shift with a cutoff of 100 MeV in comparison with experimental data and with another theoretical result, where a different halo EFT expansion has been employed [7]. One sees that our EFT expansion leads to a good description of the experimental data in the whole considered energy range, whereas the results of [7] have a less correct energy dependence beyond 2.5 MeV. Here it is worthwhile to mention that the phase shift results for both sets of solutions for the parameters $\lambda_0$ and $\lambda_1$ are practically identical.

In addition, we would like to point out that for both $\alpha\alpha$ and $\alpha n$, a Wigner bound [13] exists. It limits the cutoffs up to $\Lambda_{\alpha n}^{\text{MAX}} = 340$MeV and $\Lambda_{\alpha\alpha}^{\text{MAX}} = 230$MeV.

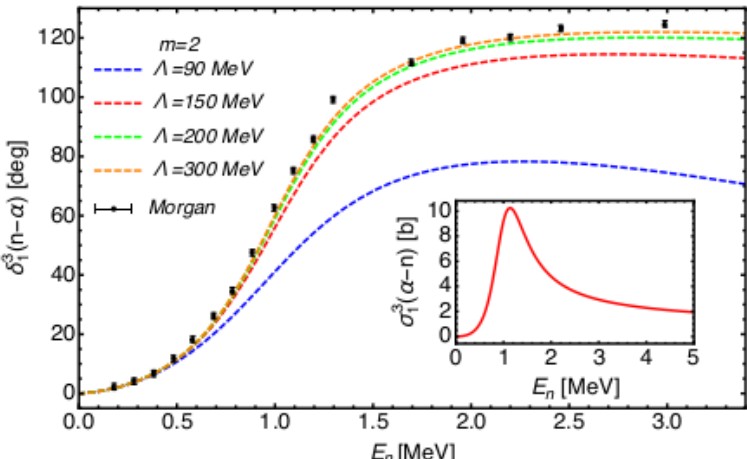

Figure 1: Phase shifts $\delta_{13}(E_n)(l=1, J=3/2)$ with experimental data from Morgan and Walter [14] and in the inset the cross-section $\sigma_{13}(E_n)$ obtained with $\Lambda = 300$MeV.

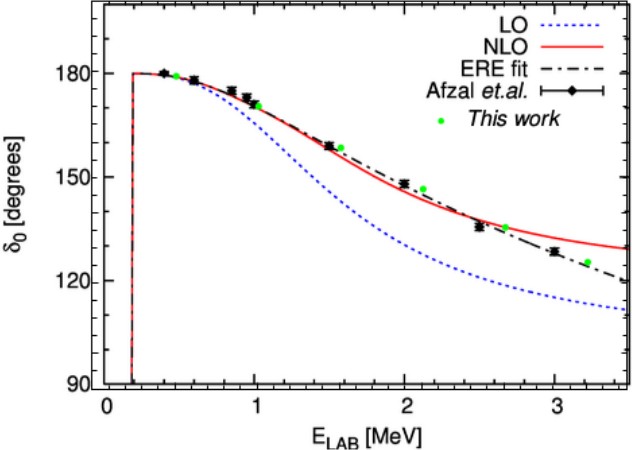

Figure 2: $\alpha\alpha$ scattering phase shift $\delta_0$ ($l=0, j=0$) with cutoff $\Lambda = 100$ MeV in comparison with experimental data from Azfal et al. [15] and with another Halo EFT calculation [7] in lowest order (LO) and Next-to-leading order (NLO). Also shown their fit using the effective range expansion formula (ERE fit).

## 3  Results

In order to obtain the ground state of the studied nuclei, we diagonalize the Hamiltonian on a nonsymmetrized hyperspherical harmonics (NSHH) basis in momentum space. The NSHH approach is based on the use of the hyperspherical harmonics basis without previous symmetrization [16–19], where the proper symmetry is then selected by means of the Casimir operator of the group of permutations of $A$ objects. This approach is very useful for fermion (boson) systems with different masses as well as for mixed boson-fermion systems, due to its extra flexibility which allows to deal with different particle systems with the same code.

Since the potentials that are used in this work are interactions born in momentum space, we chose to work with a NSHH basis in this space. HH calculations have been already carried out in momentum space [20], however with a symmetrized basis. Furthermore, in that work the momentum space HH basis was obtained from a Fourier transform of the coordinate space HH basis. Following such an approach there is an increase in complexity for the momentum space

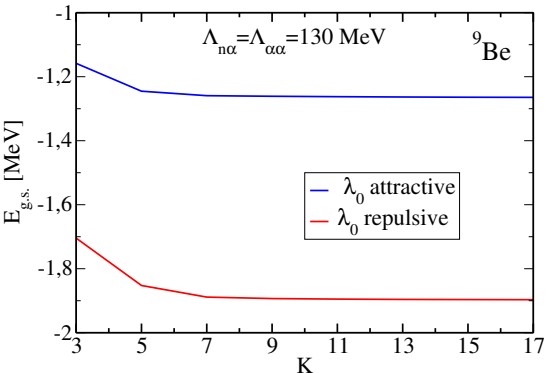

Figure 3: Convergence of $^9$Be ground state energy as a function of the HH quantum number $K$ with $\Lambda_{\alpha\alpha} = \Lambda_{\alpha n} = 130$ MeV.

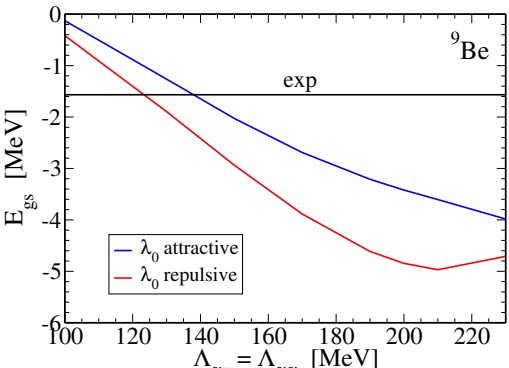

Figure 4: $^9$Be ground state energy as a function of the cutoff. Here, we use the same value for $\Lambda_{\alpha n}$ and $\Lambda_{\alpha\alpha}$.

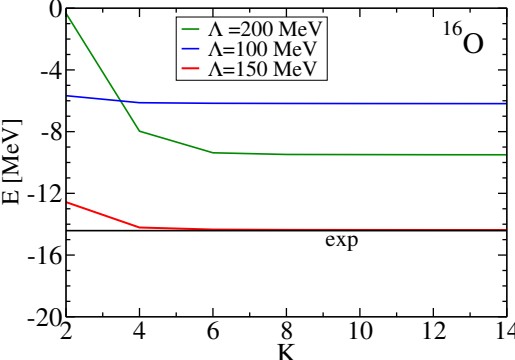

Figure 5: Convergence of $^{16}$O ground state energy as a function of the HH quantum number $K$ for different values of $\Lambda$.

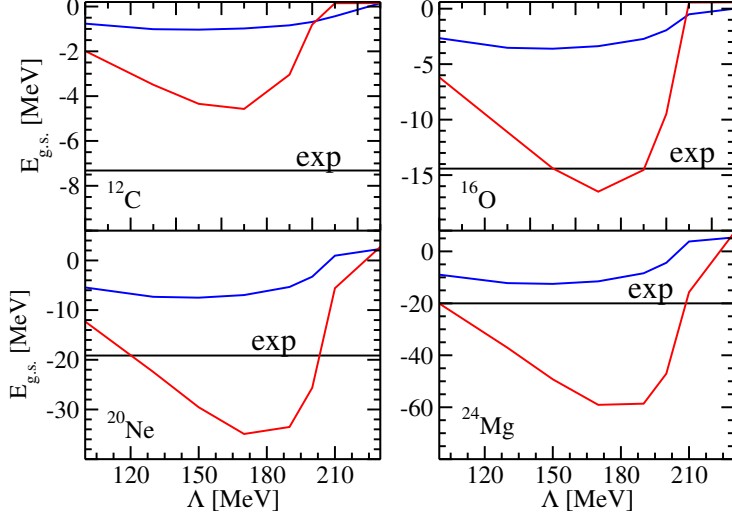

Figure 6: Cutoff dependence of ground-state energies of various $\alpha$-nuclei with HH quantum number K equal to 12 ($^{12}C$, $^{16}$O) and 8 ($^{20}$Ne, $^{24}$Mg). Blue curves: $\lambda_0$ attractive; red curves: $\lambda_0$ repulsive.

hyperradial part, since the Fourier transform of the Laguerre polynomial is a more complicated hypergeometric function that requires an enormous amount of precision in the integrations as

the required number of polynomials necessary for convergence rises. In our work, instead, a system of coordinates that is completely born in momentum space is used. It is generated in analogy to its coordinate space counterpart, namely by taking Laguerre polynomials for the hypermomentum part.

In Figure 3 we show the convergence of $^9$Be ground state energy as a function of the HH quantum number $K$, where both cutoffs are set equal to 130 MeV. The first feature that one notes is the rapid convergence. In fact, thanks to the softness of our Halo EFT potentials, one reaches quite a good convergence already at $K = 11$. Furthermore, one sees that the parameter set with a negative $\lambda_0$ leads to about 0.5 MeV less binding. In Fig. 4 we show the cutoff dependence of the ground-state energy choosing $\Lambda_{\alpha n} = \Lambda_{\alpha\alpha}$. Here one needs to take into account that the $^9$Be binding energy is given by only 1.572 MeV, that is the binding energy of the three-body $\alpha\alpha n$ cluster system, or, equivalently, the $1n$ separation energy of $^9$Be. Therefore, to obtain the total value of the $^9$Be binding energy, one has to add the binding energies of the two $\alpha$-particles. The figure shows that one obtains for some combinations of cutoff values and $\lambda_i$ solutions the experimental energy. Moreover, one notes that the $^9$Be ground-state energy exhibits a relatively strong variation, between 0 and -5 MeV, due to the cutoff value. This dependence is probably caused by the lack of a three-body force.

Now we turn to the discussion of the considered $\alpha$-nuclei. Also for these nuclei we find a rather rapid HH convergence of the various ground-state energies. In Fig. 5 we show as example $^{16}$O.

In Fig. 6 we illustrate the cutoff dependence of the ground state energies for the $\alpha$-nuclei. Again one has quite a large variation of energies. These results further support the need for a three-body force in our Halo EFT approach. It is interesting to notice, however, that, except for the case of $^{12}$C, one can find cutoff values that reproduce the experimental energy.

## 4 Conclusions

In this work we present a study of the ground-state energies of $^9$Be and various $\alpha$-nuclei with non-local $\alpha n$ and $\alpha\alpha$ interactions derived from Cluster Effective Field Theory [6–9]. The potentials are regularized by a Gaussian cutoff which takes care of ultraviolet divergences of the interaction. The potential parameters are fitted in order to reproduce a correct on-shell $T$-matrix up to the effective range order. The calculation of the various ground-state energies is carried out by diagonalizing the Hamiltonian on an NSHH basis in momentum space. We obtain in general a rather strong cutoff dependence. However, we are able to reproduce the experimental ground-state energies for selected cutoff values for all of the studied nuclei, but for $^{12}$C. The strong cutoff dependence and the case of $^{12}$C indicate the lack of three-body forces. Therefore, in future, we plan to extend our Halo EFT approach by including such many-body forces.

Another possible future project is the study of $^9$Be photodisintegration. As it was stated in the introduction, the photodisintegration of $^9$Be is an interesting reaction because it is the inverse process of $\alpha + \alpha + n \rightarrow{}^9$Be$+\gamma$, the first step in Carbon-12 production through the $\alpha\alpha n$ chain, which could be in the event of a supernova an important contribution to carbon nucleosynthesis. In order to study this process we plan to calculate the $^9$Be photoabsorption cross section via the Lorentz integral transform approach [10].

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
