# Peer review of "Beryllium-9 in Cluster Effective Field Theory"

_SciPost Physics, doi:SciPost Phys. Proc. 3, 034 (2020)_

## Round 1 · Referee Report · Daniel Phillips · 2020-1-2

Strengths

1. The approach is sound.
2. The calculations are well explained
3. The results are interesting.

Weaknesses

There are no significant weaknesses. The paper is short–as is appropriate for a conference proceedings. In the longer term I would like to see how this approach performs for excited states of these systems, e.g., the full spectrum of 9Be is interesting, and historically several states have been identified as "molecules" of a neutron and two alpha particles.

There are a couple of technical points I am puzzled about, and I have noted these in the "Requested changes".

Report

This paper develops calculations of alpha-clustered nuclei that employ separable momentum-space interactions motivated by effective field theory. The Non-Symmetrized Hyperspherical Harmonics method is used to solve the three-, four-, five-, and six-body problems with these potentials. Results are shown for the ground state energy of 9Be, 12C, 16O, 20Ne, and 24Si.

Requested changes

Several of these are simply suggestions for improvements to the English, but I also have some physics questions I would like the authors to discuss in a revised manuscript, if space permits.
1. Abstract, fourth line: should read “are fitted to reproduce the scattering length and effective range”.
2. Abstract, third-last line: I would suggest “Work is in progress on a calculation of the photodisintegration….”
3. Introduction, first line: I suggest “The idea of alpha clustering has a long history, that goes back to the 1930s.”
4. Introduction, third line: “…there is much experimental evidence for alpha clustering in nuclei”
5. Introduction, second paragraph, second line: I am not sure I would describe the energies of interest as “very low”. Energies up to almost 20 MeV should be accessible within this clustered approach, yes?
6. One of the authors of this work is a co-author of an excellent review on Halo/Cluster EFT: Hammer, Ji, Phillips, J. Phys. G 44, 103002 (2017). It would be worth citing that work, as well as the original papers already cited, when introducing the Cluster/Halo EFT concept in the text.
7. Luna and Papenbrock has recently suggested that the large scattering length found in the αα case is not the result of fine tuning, but instead is an outcome of Coulomb barrier penetration in a system with a moderately strong Coulomb repulsion (Phys. Rev. C 100, 054307 (2019)). How would this suggestion affect the power counting for the alpha-alpha system? It may also be worth comparing to Luna and Papenbrock's αα phase shifts, since they also use a finite-range regularization of the EFT potential.
8. The power counting of Eq. (2) seems to be the same as that adopted by Higa, Hammer, and van Kolck in MS Ref. [10]. A citation at this point in the text would be appropriate.
9. Do I understand correctly that the authors are only claiming LO accuracy, i.e., they have neglected NLO effects in both the α-α and α-n scattering? If so, maybe they can say that explicitly. Some readers might think “expanding up to NLO” means that NLO terms are kept.
10. The expression (8) is fine in the absence of Coulomb interactions, but–as I'm sure the authors are aware–does not apply to the α-α case, where Coulomb is present.
11. I do not think Eq. (13) is correct. Isn’t GC(+) accounted for via the denominator (p^2-k^2+i epsilon), at least if p is replaced by p’’? I therefore think that GC(+) should not appear with V inside the matrix element under the integral. Or am I missing something?
12. Why is the Non-Symmetrized Hyperspherical Harmonics basis used? Is it because these are not fermions? How is the Bose symmetry of the alpha particles imposed on the wave function of these nuclei?
13. The caption of Figs. 3 and 5 should specify that what is plotted on the x-axis is the hyper radial quantum number K. (Even though this is staled in the text.)
14. The caption of Fig. 4 should read “9Be ground state energy as a function of the cutoff…”. The same suggested change to the English applies to the opening sentence of the first full paragraph on p.6.
15. Ideally the value of K would also be quoted in the caption of Fig. 4, but since it’s in the text I suppose that is okay.
16. The first full paragraph on p.6 clarifies that the binding energy computed here does not include the binding energy of the alpha particles. That makes it the binding energy of the three-body ααn system, or, equivalently, the 1n separation energy of 9Be. Maybe these definitions of the binding energy that is being shown can be added to the text.
17. Looking at Fig. 4 & 6 it seems counter-intuitive that the repulsive αα potential leads to more binding in the 3B system. I originally wondered if my intuition is violated because this is a non-local potential. However, the one-term separable employed here seems to me to not change sign, and so be repulsive for all momenta as long as λ0>0. And the second paragraph on page 6 includes the statement “Furthermore, one sees that the parameter set with a positive λ0 leads to about 0.5 MeV less binding.”. Is it possible that the blue curves are actually for the repulsive case, in spite of the labeling in the figures?
18. Associatedly, do the authors have any insight into why the nominally attractive αα potential produces α-conjugate nuclei that are underbound, and also produces much weaker cutoff dependence than the nominally repulsive potential?

  • validity: high
  • significance: good
  • originality: good
  • clarity: high
  • formatting: good
  • grammar: reasonable

Author:  Elena Filandri  on 2020-01-31  [id 723]

(in reply to Report 1 by Daniel Phillips on 2020-01-02)
Category:
answer to question

We thank the Referee for his suggestions for improving the English and for raising a number of important points. According to the Referee’s comments, the paper has been revised. We made the suggested changes ( points 1,2,3,4,5,10,11,13 and 14 of ”Requested changes”) and added new citations (points 6 and 8) to improve the paper contents. Concerning his other points we give a more detailed reply in the following.

  1. Luna and Papenbrock has recently suggested that the large scattering length found in the αα case is not the result of fine tuning, but instead is an outcome of Coulomb barrier penetration in a system with a moderately strong Coulomb repulsion (Phys. Rev. C 100, 054307 (2019)). How would this suggestion affect the power counting for the alpha-alpha system? It may also be worth comparing to Luna and Papenbrock’s αα phase shifts, since they also use a finite-range regularization of the EFT potential.

Unfortunately, we didn’t have enough time to make a detailed comparison with this work but we planned to do so in future.

  1. Do I understand correctly that the authors are only claiming LO accuracy, i.e., they have neglected NLO effects in both the αα and αn scattering? If so, maybe they can say that explicitly. Some readers might think ”expanding up to NLO” means that NLO terms are kept.

With the adopted power counting, in the αn interaction case, the scattering length a 1 and the effective range r 1 contribute to the leading order (LO), there are no contributions at the next-to-leading order (NLO) and the shape parameter P 1 is next-to-next-to leading order (N2LO). In the case of αα interaction a 0 and r 0 give contributions to the LO, there are no contributions at the NLO and the shape parameter P 0 which in Ref.10 contributes in NLO in our power counting is of a higher order. This because Higa et al. expanded the phase shift around k R /k c , where k R is the alpha-alpha resonance momentum and k c is the Coulomb momentum. On the contrary we found that it is not necessary to expand the Coulomb corrected unitary term ( H(η) in the paper). One can expand around k R /M hi instead, where M hi is the breakdown scale for short-range interaction. If we expand H(η) at k/k c , we will obtain ( a + P ) k 4 , where P is shape parameter but a is from the H(η) expansion. In fact, a is much larger than P, which dominate at the k 4 contribution. Therefore, the contributions from the shape parameter actually enters at higher orders than NLO. This is reflected by the result on Fig.2, our work included only a 0 and r 0 and already agrees better with the ERE results than Higa’s with shape parameter. This is purely because their expansion is only accurate at very small momentum. We clarified on p.2 ”we perform an EFT expansion up to the effective range order ” and then at p. 3: ”With the adopted power counting, in the αn interaction case, the scattering length a 1 and the effective range r 1 contribute to the leading order (LO), there are no contributions at the next-to-leading order (NLO) and the shape parameter P 1 is next-to-next-to leading order (N2LO). In the case of αα interaction a 0 and r 0 give contributions to the LO, there are no contributions at the NLO and the shape parameter P 0 is of a higher order.”

  1. Why is the Non-Symmetrized Hyperspherical Harmonics basis used? Is it because these are not fermions? How is the Bose symmetry of the alpha particles imposed on the wave function of these nuclei?

We clarified this point on p. 5 ”The NSHH approach is based on the use of the hyperspherical harmonics basis without previous symmetrization (see Ref. 14-16), where the proper symmetry is then selected by means of the Casimir operator of the group of permutations of A objects. This approach is very useful for fermion (boson) systems with different masses as well as for mixed boson-fermion systems, due to its extra flexibility which allows to deal with different particle systems with the same code. ”

  1. The first full paragraph on p.6 clarifies that the binding energy computed here does not include the binding energy of the alpha particles. That makes it the binding energy of the three-body ααn cluster system, or, equivalently, the 1n separation energy of 9Be. Maybe these definitions of the binding energy that is being shown can be added to the text.

We added these definitions (p. 6 ”Here one needs to take into account that the 9 Be binding energy is given by only 1.572 MeV, that is the binding energy of the three-body ααn system, or, equivalently, the 1n separation energy of 9 Be. Therefore, to obtain the total value of the 9 Be binding energy, one has to add the binding energies of the two α-particles”).

  1. Looking at Fig. 4 & 6 it seems counter-intuitive that the repulsive αα potential leads to more binding in the 3B system. I originally wondered if my intuition is violated because this is a non-local potential. However, the one-term separable employed here seems to me to not change sign, and so be repulsive for all momenta as long as λ 0 >0. And the second paragraph on page 6 includes the statement ”Furthermore, one sees that the parameter set with a positive λ 0 leads to about 0.5 MeV less binding.”. Is it possible that the blue curves are actually for the repulsive case, in spite of the labeling in the figures?
  2. Associatedly, do the authors have any insight into why the nominally attractive αα potential produces α-conjugate nuclei that are underbound, and also produces much weaker cutoff dependence than the nominally repulsive potential?

When we solve the Lippmann-Schwinger equation to find the potential constants we get a second order equation with two pairs of solutions, one with a positive λ 0 and negative λ 1 and one with a negative λ 0 and positive λ 1 . It is worth pointing out that both sets of solutions generate an attractive potential. Moreover, even if in this preliminary phase we studied both solutions pairs for the potential constants, we think that the set with negative λ 0 (therefore λ 0 attractive) is preferable because less cutoff dependent. In the text we have corrected the error ”Furthermore, one sees that the parameter set with a positive λ 0 leads to...” → "Furthermore, one sees that the parameter set with a negative λ 0 leads to ..” and we clarified the fact that our potential is attractive for both pairs of solutions p. 4 ”.. one with a positive λ 0 and negative λ 1 and one with a negative λ 0 and positive λ 1 . Later we will call the first solution λ 0 repulsive and the second one λ 0 attractive, it is worth pointing out that both sets of solutions generate an attractive potential between the particles”

Anonymous on 2020-02-04  [id 726]

(in reply to Elena Filandri on 2020-01-31 [id 723])
Category:
remark

We confirm that we made all modifications of the MS as illustrated in our response to the Referee, we hope the revised version of MS will be available in the next few days. We prefer not to make a citation of the work of Luna and Papenbrock since here we can not yet comment on it , while, as already said, we want to make further investigations on our side and, on that occasion, make according comments.

Daniel Phillips  on 2020-02-03  [id 724]

(in reply to Elena Filandri on 2020-01-31 [id 723])
Category:
remark

The authors have dealt with almost all the issues I raised in my original report. I would request that they add a citation to Luna & Papenbrock's paper, but this is not essential.

I am not completely sure that my last question has been answered, since I can't seem to view the revised MS. But what the authors' say in their response makes sense, and I am happy to trust that they have clarified sufficiently.

---

## Editorial Decision

published